# Optimizing hierarchical tree dissection parameters using historic epidemiologic data as 'ground truth'

David Jacobson[1,2]*, Joel Barratt[1]

1 Parasitic Diseases Branch, Division of Parasitic Diseases and Malaria, Centers for Disease Control and Prevention, Atlanta, Georgia, United States of America, 2 Oak Ridge Institute of Science and Education, Oak Ridge, Tennessee, United States of America

* quh7@cdc.gov

**Data Availability Statement:** All relevant data are within the manuscript and its Supporting information files.

**Funding:** The authors received no specific funding for this work.

## Abstract

Hierarchical clustering of pathogen genotypes is widely used to complement epidemiologic investigations of outbreaks. Investigators must dissect trees to obtain genetic partitions that provide epidemiologists with meaningful information. Statistical approaches to tree dissection often require a user-defined parameter to predict the optimal partition number and augmenting this parameter can drastically impact resultant partition memberships. Here, we demonstrate how to optimize a given tree dissection parameter to maximize accuracy irrespective of the tree dissection method used. We hierarchically clustered 1,873 genotypes of the foodborne pathogen *Cyclospora* spp., including 587 possessing links to historic outbreaks. We dissected the resulting tree using a statistical method requiring users to select the value of a 'stringency parameter' (*s*), with a recommended value of 95% to 99.5%. We dissected this hierarchical tree across *s*-values from 94% to 99.5% (at increments of 0.25%), to identify a value that maximized partitioning accuracy, defined as the degree to which genetic partitions conform to known epidemiologic groupings. We show that *s*-values of 96.5% and 96.75% yield the highest accuracy (> 99.9%) when clustering *Cyclospora* sp. isolates with known epidemiologic linkages. In practice, the optimized *s*-value will generate robust genetic partitions comprising isolates likely derived from a common food source, even when the epidemiologic grouping is not known prior to genetic clustering. While the *s*-value is specific to the tree dissection method used here, the optimization approach described could be applied to any parameter/method used to dissect hierarchical trees.

## Introduction

Hierarchical clustering is widely used in the field of molecular epidemiology to detect groups of genetically related pathogen isolates. However, an important limitation of hierarchical clustering is that hierarchical clusters are nested, meaning that small clusters comprising closely related isolates exist within larger clusters that get progressively larger as genetic relationships become increasingly distant. Consequently, investigators must dissect hierarchical trees into

**Competing interests:** The authors have declared that no competing interests exist.

discrete genetic groupings (i.e., partitions) to facilitate prioritization of discrete genetic groups for subsequent epidemiologic investigation. Usually, the value of some tree-dissection parameter (e.g., a SNP distance threshold) is empirically selected by investigators to facilitate tree dissection, hopefully yielding partitions where all (or most) grouped isolates are representatives of the same strain [1–3]. In epidemiologic contexts, the objective is always to select a parameter value for tree dissection that groups isolates with a high likelihood of belonging to the same strain, and thus, have a high probability of being associated with a common source.

Various statistical methods exist that can be used to guide tree dissection by predicting an optimal partition number [4], yet these methods usually require users to select a value for one or more input parameters that can have a significant impact on the resulting partition memberships. As such, the value of any user-defined parameter for tree dissection should be set with careful consideration. Values yielding too few partitions can link dissimilar isolates together, making it difficult to identify suspected food vehicles. Alternatively, values yielding too many partitions may separate genetically similar isolates, causing outbreaks to be overlooked. Empirical selection of a user-defined input parameter value during tree dissection may yield accurate partitions, particularly for pathogens for which a large volume of robust historical molecular epidemiologic data is available. This is because historical genetic data can inform molecular epidemiologists of how genetically similar isolates of the same strain typically are; however, for many human parasites, including the foodborne parasite *Cyclospora* spp. [5], historic knowledge of circulating strains may be limited or absent, and the concept of what constitutes a strain may be complicated by sexual reproduction [6].

The intersect of an epidemiologically-defined cluster and its analogous genetic cluster will ideally be approaching 100%: this principle forms the basis of molecular epidemiology [7]. For example, a recently described tool based on multi-locus-sequence-typing (MLST) and hierarchical clustering for genotyping *Cyclospora* spp., generally displays approximately 90% concordance with epidemiologic data [8, 9]. However, routine *Cyclospora* spp. genotyping only began in the United States in 2018 [8], so the volume historic molecular data available for this pathogen is limited compared to available data for foodborne bacterial pathogens such *E. coli* O157 or *Salmonella* [1, 10, 11]. For the latter two bacterial pathogens, data on intra-strain genetic variation is available to inform selection of certain partitioning thresholds such as species-specific SNP-difference threshold [1, 10, 11].

Alternatively, the methods used for identification of discrete partitions within hierarchically clustered *Cyclospora* spp. data requires continued optimization. Given the current lack of historic 'strain' information for *Cyclospora* spp., we propose here that historic outbreak-linked genotypes could be used to optimize tree dissection parameters, by maximizing the degree to which genetic partitions conform to known epidemiologic groupings (i.e., maximizing partitioning accuracy against an epidemiologic gold standard). Importantly, the outbreak-linked, 'gold standard', genotypes used in this optimization must be confidently linked to an epidemiologic cluster, as speculative epidemiologic groupings may misrepresent true algorithmic performance. Subsequently, following optimization, these historic genotypes could be hierarchically clustered alongside genotypes from isolates of unknown linkage. On partitioning of the resultant hierarchical tree using optimized parameter values, resultant partitions comprising isolates with unknown linkage have a high likelihood of being derived from a common source and should be prioritized for epidemiologic follow-up. Optimization with gold standard epidemiologically linked genotypes has already proved successful in identifying high performing genetic distance calculation algorithms to use in *Cyclospora* spp. genotyping. We previously clustered matrices generated using common distance calculation approaches (e.g., Jaccard, Bray-Curtis, Manhattan, and Euclidean) as well as novel haplotype-based algorithms designed for sexually reproducing parasites (Barratt's heuristic and Plucinski's Bayesian), with

Barratt's heuristic outperforming all other methods when evaluating how accurately genetic clusters reflect gold standard epidemiologic clusters [12], as we propose to do here.

In a recent study, dissection of hierarchically clustered *Cyclospora* spp. MLST data to identify discrete partitions comprising closely related isolates, was performed using a statistical framework that requires selecting a value for the user-defined 'stringency' parameter [4]. In that study [4], we recommended that the stringency parameter be set to a value above 95% and below 100%, though we justified the use of the maximum recommended *s*-value of 99.5% to dissect a hierarchically clustered dataset of more than 1,000 *Cyclospora* spp. MLST genotypes [4]. Setting the stringency to 99.5% resulted in the delimitation of genetic partitions where 90.8% of epidemiologically linked isolates were also linked genetically (i.e., 90.8% sensitivity) [4]. We also advised that users should consider optimizing the stringency parameter (*s*) to maximize performance, though specific details on how this may be achieved were not provided [4]. Therefore, the aim of this study was to demonstrate how a given tree dissection parameter–in this case, the value of the stringency parameter—can be optimized using historic epidemiologic data to improve tree dissection accuracy. Ultimately, we show that compared to when tree dissection parameter values are empirically selected, optimization of parameters in the way described does result in genetic partitions that more accurately reflect the epidemiologic linkage of clustered genotypes.

## Materials and methods

### Genotyping data

We utilized a publicly available MLST dataset for *Cyclospora* spp. generated by the United States (U.S.) Centers for Disease Control and Prevention (CDC), the Public Health Agency of Canada, and certain U.S. State public health departments, as part of ongoing *Cyclospora* spp. genotyping performed during 2018, 2019, 2020, and 2021 [8, 9, 13–18]. To maximize the diversity of isolates included this analysis, we also included genotypes from persons who became infected in China and Indonesia, and from persons presenting with cyclosporiasis in the UK after returning from travel. Briefly, this dataset comprised 1,873 *Cyclospora* sp. genotypes. These isolates had been sequenced at eight markers as previously described [8, 9, 18], including six nuclear markers and two mitochondrial markers. Illumina data from these isolates were accessed under NCBI BioProject Number PRJNA578931. Each isolates' genotype had been ascertained using bioinformatic workflows previously described [8].

### Epidemiologic information

Epidemiologic information for a subset of these 1,873 genotypes was collected prior to this study through Cyclosporiasis National Hypothesis Generating Questionnaires (CNHGQ) during routine US public health surveillance. Each CNHGQ included information on a case-patient's food consumption history during a two-week period before becoming ill. Using this information, 587 isolates included in this analysis had been confidently linked to an outbreak or event that occurred in the USA, for which more than one isolate was genotyped (Table 1). Genotypes possessing clear epidemiologic links represented a reference for expected (i.e., 'ground truth') clustering outcomes when assessing clustering performance (see below). Isolates that could not be linked confidently to an outbreak cluster were designated as possessing "unknown epidemiologic linkage". Isolates in this "unknown" category also included all isolates from outside the USA as CNHGQs were not collected for cyclosporiasis patients outside the USA.

**Table 1. List of epidemiologic clusters and their size.**

| Epidemiologic Cluster | Epi-Cluster Number[a] | Size | Epidemiologic summary |
|---|---|---|---|
| Pre-packaged salad mix 2020_001 (2020) | 01 | 132 | Contaminated salad product identified; precise vehicle uncertain. |
| Vendor A (2018) | 02 | 94 | Contaminated salad product identified, though precise vehicle uncertain. |
| Pre-packaged salad mix 2020_003 (2020) | 03 | 76 | Contaminated salad product identified; precise vehicle uncertain. |
| Vendor B (2018) | 04 | 63 | Contaminated vegetable tray product identified; precise vehicle uncertain. |
| Distributor A–Type 17 (2019) | 05 | 41 | Vehicle (an herb) identified. |
| 2021 July Romaine 1–Type 9 | 06 | 22 | Vehicle (a leafy green) identified. |
| Distributor A–Type 3 (2019) | 07 | 17 | Vehicle (an herb) identified. |
| 2021 August Butter lettuce 1 | 08 | 13 | Vehicle (a leafy green) identified. |
| Distributor A–Type 18 (2019) | 09 | 13 | Vehicle (an herb) identified. |
| Restaurant A (2019) | 10 | 13 | Vehicle (an herb) identified. |
| Distributor A–Type 1 (2019) | 11 | 12 | Vehicle (an herb) identified. |
| Restaurant D (2019) | 12 | 12 | Vehicle/product uncertain |
| Restaurant B (2019) | 13 | 11 | Vehicle (a leafy green) identified. |
| Tennessee/Georgia/Virginia Mexican-style restaurant / cilantro sub-cluster (2020) | 14 | 10 | Vehicle (an herb) identified. |
| 2021 July Romaine 1–Type 1 | 15 | 9 | Vehicle (a leafy green) identified. |
| Temporospatial Cluster A (2018) | 16 | 8 | Vehicle/product uncertain |
| Prepackaged salad 002 (2020) | 17 | 7 | Contaminated salad product identified; precise vehicle uncertain. |
| Restaurant C (2019) | 18 | 6 | Herb spread product identified; precise vehicle uncertain |
| Supplier X–Restaurants A and B (Herb 1) Associated Cluster (2018) | 19 | 6 | Vehicle (an herb) identified. |
| 2021 July Romaine 1–Type 5 | 20 | 5 | Vehicle (a leafy green) identified. |
| 2021 TN Restaurant 1 (TN21-022) | 21 | 5 | Vehicle/product uncertain |
| Salad Chain A–2020_025 (2020) | 22 | 4 | Vehicle/product uncertain |
| 2021 Connecticut Bridal 1 | 23 | 2 | Vehicle/product uncertain |
| 2021 Florida Italian-style restaurant | 24 | 2 | Vehicle/product uncertain |
| North Dakota Market Salad Cluster | 25 | 2 | Contaminated salad product identified, though precise vehicle uncertain. |
| Restaurant C (Herb 2) Associated Cluster (2018) | 26 | 2 | Vehicle (an herb) identified. |

Full name of epidemiologic clusters included in this manuscript's analysis. The size column indicates the number of cases with a successfully genotyped isolate in each epidemiologic cluster. This number is independent of genetic partition membership.

[a]Epi-Cluster Number (E.C.N) is an arbitrary number applied to epidemiologic cluster to allow for easy reference.

### Distance calculation and partition number selection

A pairwise distance matrix was calculated from these *Cyclospora* spp. genotypes using Barratt's heuristic definition of genetic distance as previously described [3, 19, 20]. This matrix was hierarchically clustered using Ward's method implemented via the agnes function in the R package 'cluster' [21]. Next, we applied Plucinski and Barratt's framework as previously described [4] to dissect the resulting hierarchical tree into a *k* number of discrete partitions across 23 different stringency values (*s*-values): those ranging from 94% to 99.5%, at intervals of 0.25%. The number of discrete partitions (*k*) predicted using each of these 23 *s*-values was recorded. We subsequently dissected the hierarchical tree into the number of partitions (*k*) predicted for each *s*-value using the cutree R function [22]. The partition memberships

resulting from each of these 23 tree dissection iterations was used to assess partitioning performance for each of the corresponding 23 *s*-values. All hierarchical trees in this manuscript were generated using ggtree in R [23].

## Assessment of partitioning performance

For each of the 23 *s*-values tested, we classified clustering results obtained for each genotype as either a true positive (TP), false positive (FP), true negative (TN), or false negative (FN), using the definitions described below. From these classifications we calculated various performance metrics including sensitivity, specificity, positive predictive value (PPV), negative predictive value (NPV), and accuracy, as previously described [8]. The calculations were weighted by the ratio of genotyped isolates in each epidemiologic cluster to the total number of genotyped isolates with epidemiologic links (n = 587) so that larger epidemiologic clusters (i.e., with more genotyped isolates) would contribute more to the final values. Given that accuracy is a measure of proximity of results from the true value, we proposed that the optimal stringency setting would be the value of *k* that results in maximum accuracy, as determined by the equation:

$$Accuracy = \frac{(TP + TN)}{(TP + TN + FP + FN)}$$

After identifying the stringency setting that maximized accuracy, we assessed the discriminatory power of obtained using this setting by calculating Simpson's index of diversity (*D*) as described elsewhere [7]. The value of *D* was determined by:

$$D = 1 - \left( \frac{1}{N(N-1)} \times \sum_{J=1}^{S} n_j \left( n_j - 1 \right) \right)$$

where *N* is the total number of isolates (n = 1,873), *S* is the number of partitions (i.e., equal to *k*), and $n_j$ represents the number of isolates within the *j*th partition. *D* is calculated with all isolates, not just those with epidemiological linkages. Simpson's index assesses a method's ability to distinguish between unrelated strains sampled randomly from a given species [7], where values of *D* close to 1.0 generally indicate good discriminatory power. We therefore considered this an indicator of whether the optimal stringency value (i.e., the one that maximizes accuracy) also provides useful strain discrimination.

## Classification of epidemiologically-linked isolates after clustering

To compute partitioning accuracy, each of the 587 isolates with epidemiologic links were classified as a TP, TN, FP, or FN based on whether they were correctly assigned to the same partition as their epidemiologically-linked partners or not. Previous investigations showed that most epidemiologically-linked isolates included in this analysis possess a similar genetic signature [8, 9, 18]. Therefore, each epidemiologic cluster would have a partition number (i.e., a genetic cluster) to which the majority of its epidemiologically-linked isolates would be assigned. For the purposes of classification, we refer to this as the 'mode' partition number for an epidemiologic cluster. True positives would comprise isolates that were correctly assigned to the mode partition number for their epidemiologic cluster. Next, if we consider a fictitious epidemiologic cluster called "Outbreak A", true negatives for the "Outbreak A" cluster would include all isolates from Outbreaks X, Y, and Z that were not assigned to the mode partition for outbreak A. False negatives would include isolates that were not assigned to the mode partition number for their epidemiologic cluster. False positives would include isolates with a particular epidemiologic linkage that were assigned to a different partition to that of their

epi-linked partners, and to a partition alongside isolates with a different epidemiologic linkage. Importantly, isolates belonging to different epidemiologic clusters can share the same mode partition number (i.e., unrelated outbreaks caused by the same strain). Therefore, isolates with the same mode partition number but possess a distinct epidemiologic linkage were not classified as false positive linkages for the purposes of our analysis. The classifications were performed for each epidemiologic cluster separately and the sum of all TP, TN, FP, and FN classifications for each epidemiologic cluster was used to compute an overall value of clustering accuracy for each stringency setting used, in addition to other performance metrics.

## Ethics

Ethics approval for the use of clinical specimens was reviewed by the CDC Center for Global Health Human Research Protection Office under project determination number 2018–123. The need for patient informed consent was waived because the specimens were de-linked from any personal identifiers prior to submission to CDC.

## Results

Stringency values of 96.5% and 96.75% produced identical performance results (Table 2) and were established as optimal for partitioning our hierarchically clustered dataset. All $s$-values $\geq$ 96.5% resulted in partitions with zero false positive links, yielding a specificity and PPV of 1 (Table 2). Conversely, all values $\leq$ 96.75% resulted in partitions with the fewest false negatives (n = 6), which maximized sensitivity and NPV (Table 2). Consequently, the optimal $s$-values of 96.5% and 96.75% meant we minimized both false positives and false negatives,

**Table 2. Clustering performance for each stringency ($s$) value.**

| Stringency values ($s$) | 94 through 94.5 | 94.75 | 95 | 95.25 | 95.5 | 95.75 and 96 | 96.25 | 96.5 and 96.75 | 97 through 97.75 | 98 | 98.25 | 98.5 | 98.75 | 99 | 99.25 | 99.5 |
|---|---|---|---|---|---|---|---|---|---|---|---|---|---|---|---|---|
| Predicted partition number ($k$) | 21 | 23 | 24 | 25 | 26 | 27 | 28 | 30 | 31 | 32 | 35 | 40 | 48 | 57 | 63 | 67 |
| True positives | 581 | 581 | 581 | 581 | 581 | 581 | 581 | 581 | 571 | 568 | 568 | 568 | 556 | 555 | 554 | 548 |
| True negatives | 13921 | 13921 | 13921 | 13921 | 13922 | 13969 | 13969 | 13994 | 13994 | 13994 | 13994 | 13994 | 13994 | 13994 | 13994 | 13994 |
| False Positives | 73 | 73 | 73 | 73 | 72 | 25 | 25 | 0 | 0 | 0 | 0 | 0 | 0 | 0 | 0 | 0 |
| False negatives | 6 | 6 | 6 | 6 | 6 | 6 | 6 | 6 | 16 | 19 | 19 | 19 | 31 | 32 | 33 | 39 |
| Sensitivity | 0.9898 | 0.9898 | 0.9898 | 0.9898 | 0.9898 | 0.9898 | 0.9898 | 0.9898 | 0.9727 | 0.9676 | 0.9676 | 0.9676 | 0.9472 | 0.9455 | 0.9438 | 0.9336 |
| Specificity | 0.9948 | 0.9948 | 0.9948 | 0.9948 | 0.9949 | 0.9982 | 0.9982 | 1.0000 | 1.0000 | 1.0000 | 1.0000 | 1.0000 | 1.0000 | 1.0000 | 1.0000 | 1.0000 |
| NPV | 0.9996 | 0.9996 | 0.9996 | 0.9996 | 0.9996 | 0.9996 | 0.9996 | 0.9996 | 0.9989 | 0.9986 | 0.9986 | 0.9986 | 0.9978 | 0.9977 | 0.9976 | 0.9972 |
| PPV | 0.8884 | 0.8884 | 0.8884 | 0.8884 | 0.8897 | 0.9587 | 0.9587 | 1.0000 | 1.0000 | 1.0000 | 1.0000 | 1.0000 | 1.0000 | 1.0000 | 1.0000 | 1.0000 |
| Accuracy | 0.9946 | 0.9946 | 0.9946 | 0.9946 | 0.9947 | 0.9979 | 0.9979 | 0.9996 | 0.9989 | 0.9987 | 0.9987 | 0.9987 | 0.9979 | 0.9978 | 0.9977 | 0.9973 |
| Number of clusters (only epi-linked isolates) | 15 | 15 | 15 | 15 | 15 | 16 | 16 | 17 | 17 | 17 | 17 | 17 | 17 | 17 | 17 | 17 |
| Simpson's D | 0.9100 | 0.9105 | 0.9109 | 0.9110 | 0.9114 | 0.9128 | 0.9131 | 0.9210 | 0.9247 | 0.9251 | 0.9253 | 0.9268 | 0.9280 | 0.9285 | 0.9310 | 0.9335 |

Cells color relates to optimal values, were red indicates values that are sub-optimal and dark green represents values that are optimal. Optimal $s$-values (those that maximize accuracy) are shaded in gray and surrounded by a dark box.

[a]There were 26 epidemiologic clusters. Given that some outbreak clusters were caused by the same strain (Table 3), the epidemiologic clusters were distributed across 15 to 17 genetic partitions depending on the stringency value used. Many genetic partitions were not associated with epidemiologic data, which is also why epidemiologically-linked isolates are only spread across 15 to 17 partitions (stringency dependent).

Table 3. Impact of stringency setting on the partition (*k*) membership of genotypes linked to various epidemiologic clusters.

| Stringency values (*s*) | 94 through 94.5 | 94.75 | 95 | 95.25 | 95.5 | 95.75 and 96 | 96.25 | 96.5 and 96.75 | 97 through 97.75 | 98 | 98.25 | 98.5 | 98.75 | 99 | 99.25 | 99.5 |
|---|---|---|---|---|---|---|---|---|---|---|---|---|---|---|---|---|
| Predicted partition number (*k*) | 21 | 23 | 24 | 25 | 26 | 27 | 28 | 30 | 31 | 32 | 35 | 40 | 48 | 57 | 63 | 67 |
| Mode partition numbers for each stringency value are shown below | | | | | | | | | | | | | | | | |
| Temp. Cluster A (2018)–16[a] | 3 | 3 | 3 | 3 | 3 | 27 | 28 | 30 | 31 | 32 | 35 | 40 | 48 | 57 | 63 | 67 |
| 2021 FL Italian-style res.–24[a] | 3 | 3 | 3 | 3 | 3 | 3 | 3 | 19 | 20 | 21 | 21 | 22 | 23 | 24 | 25 | 26 |
| Salad Chain A (2020)–22[a] | 3 | 3 | 3 | 3 | 3 | 3 | 3 | 3 | 3 | 3 | 3 | 3 | 3 | 3 | 3 | 3 |
| Dist. A–Type 3 (2019)–07[a] | 3 | 3 | 3 | 3 | 3 | 3 | 3 | 3 | 3 | 3 | 3 | 3 | 3 | 3 | 3 | 3 |
| 2021 August Butter lettuce 1–08 | 12 | 12 | 12 | 12 | 12 | 12 | 12 | 12 | 15 | 16 | 16 | 17 | 18 | 19 | 20 | 20 |
| ND Market Salad Cluster–25 | 12 | 12 | 12 | 12 | 12 | 12 | 12 | 12 | 15 | 16 | 16 | 17 | 18 | 19 | 20 | 20 |
| Salad mix 2020_001 (2020)–01 | 12 | 12 | 12 | 12 | 12 | 12 | 12 | 12 | 15 | 16 | 16 | 17 | 18 | 19 | 20 | 20 |
| Prepackaged salad 002 (2020)–17 | 11 | 11 | 11 | 11 | 11 | 11 | 11 | 11 | 11 | 12 | 12 | 12 | 12 | 12 | 12 | 12 |
| Res. A (2019)–10 | 11 | 11 | 11 | 11 | 11 | 11 | 11 | 11 | 11 | 12 | 12 | 12 | 12 | 12 | 12 | 12 |
| Supplier X–(Herb 1) - 19 | 11 | 11 | 11 | 11 | 11 | 11 | 11 | 11 | 11 | 12 | 12 | 12 | 12 | 12 | 12 | 12 |
| 2021 CT Event 1–23 | 5 | 5 | 5 | 5 | 5 | 5 | 5 | 5 | 5 | 5 | 5 | 5 | 5 | 5 | 5 | 5 |
| 2021 July Romaine 1–Type 5–20 | 5 | 5 | 5 | 5 | 5 | 5 | 5 | 5 | 5 | 5 | 5 | 5 | 5 | 5 | 5 | 5 |
| 2021 TN Res. 1 (TN21-022)–21 | 5 | 5 | 5 | 5 | 5 | 5 | 5 | 5 | 5 | 5 | 5 | 5 | 5 | 5 | 5 | 5 |
| Salad mix 2020_003 (2020)–03 | 5 | 5 | 5 | 5 | 5 | 5 | 5 | 5 | 5 | 5 | 5 | 5 | 5 | 5 | 5 | 5 |
| 2021 July Romaine 1–Type 1–15 | 1 | 1 | 1 | 1 | 1 | 1 | 1 | 1 | 1 | 1 | 1 | 1 | 1 | 1 | 1 | 1 |
| Dist. A–Type 17 (2019)–05 | 1 | 1 | 1 | 1 | 1 | 1 | 1 | 1 | 1 | 1 | 1 | 1 | 1 | 1 | 1 | 1 |
| 2021 July Romaine 1–Type 9–06 | 9 | 9 | 9 | 9 | 9 | 9 | 9 | 9 | 9 | 10 | 10 | 10 | 10 | 10 | 10 | 10 |
| Dist. A–Type 1 (2019) - 11 | 13 | 13 | 13 | 13 | 19 | 19 | 19 | 20 | 21 | 22 | 22 | 35 | 39 | 41 | 44 | 46 |
| Dist. A–Type 18 (2019)–09 | 15 | 15 | 15 | 15 | 15 | 15 | 15 | 15 | 16 | 17 | 17 | 18 | 29 | 30 | 32 | 33 |
| Res. B (2019)–13 | 7 | 7 | 7 | 7 | 7 | 7 | 7 | 7 | 7 | 8 | 8 | 8 | 8 | 8 | 8 | 8 |
| Res. C (2019)–18 | 2 | 2 | 2 | 2 | 2 | 2 | 2 | 2 | 2 | 2 | 2 | 2 | 2 | 2 | 2 | 2 |
| Res. C (Herb 2) 2018–26 | 21 | 22 | 23 | 24 | 25 | 25 | 26 | 27 | 28 | 29 | 30 | 32 | 34 | 35 | 37 | 38 |
| Res. D (2019)–12 | 18 | 19 | 20 | 20 | 21 | 21 | 21 | 22 | 23 | 24 | 24 | 26 | 27 | 28 | 30 | 31 |
| Mexican-style res. / cilantro sub-cluster (2020)–14 | 16 | 16 | 16 | 16 | 16 | 16 | 16 | 16 | 17 | 18 | 18 | 19 | 20 | 21 | 22 | 22 |
| Vendor A (2018)–02 | 19 | 20 | 21 | 21 | 22 | 22 | 22 | 23 | 24 | 25 | 25 | 27 | 38 | 39 | 42 | 44 |
| Vendor B (2018)–04 | 4 | 4 | 4 | 4 | 4 | 4 | 4 | 4 | 4 | 7 | 7 | 7 | 7 | 7 | 7 | 7 |

Colored rows indicate multiple outbreaks caused by the same strain, as evidenced by the fact that at all values of *k* the isolates associated with these different outbreaks remained genetically linked. Therefore, genetic linkage of isolates from unrelated outbreaks shaded the same color was not considered a false-positive linkage in this study because the causative strains were the same based on the genotyping scheme employed. Res. = restaurant, Dist. = Distributor, Temp. = temporospatial.

[a]Four discrete epidemiologically-defined clusters that were originally assigned to the same genetic partition at lower stringency values, yet separate into three distinct genetic partitions at higher stringency values

which yielded partitions with the highest accuracy (99.96%). At the optimal *s*-values, the 1,873 genotyped isolates were distributed across 30 partitions (i.e., *k* = 30) (Tables 2 and 3). The optimal stringency setting also maximized the number of partitions that isolates with epidemiologic links (n = 587) were distributed across (17 partitions, Table 2). Sub-optimal accuracy was observed at an *s*-value of 99.5%, which maximized the number of false negatives (n = 39) as a consequence of dividing many linked isolates across different partitions (Table 2). Conversely, an *s*-value of 94% (the minimum value evaluated) yielded 73 false positives, due to assignment of many unrelated isolates to the same partition.

The higher number of false positives at lower stringency values is a consequence of isolates from different epidemiologic clusters being assigned to the same partition. Specifically, this

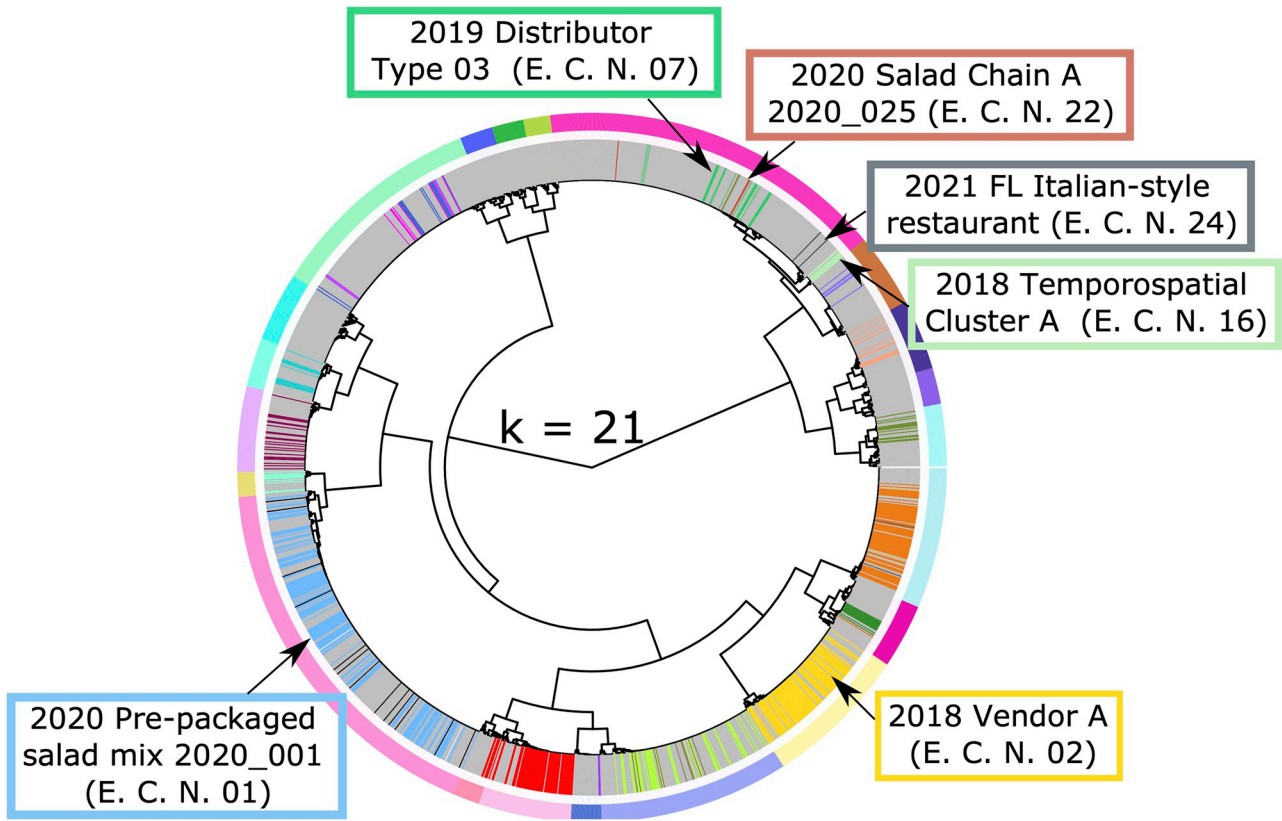

**Fig 1. Impact of minimum stringency (*s*) value on genetic linking of epidemiologically-linked isolates.** The hierarchical tree generated from our distance matrix was dissected into the minimum value of *k* (*k* = 21) predicted using the 94% *s*-value. The outer circle of colored bars indicates the boundary between each partition predicted and the inner circle of colored bars represents the epidemiologic linkage of various isolates, where each bar color represents a distinct epidemiologic cluster (grey represents isolates of unknown epidemiologic linkage). Epidemiological clusters of interest are labeled in the colored boxes. (A) At *k* = 21, we observe that the labeled epidemiologic clusters on the top right of the tree all belong to a single genetic partition, indicating that different epidemiologic clusters are being unnecessarily grouped into a single genetic partition.

included isolates from cyclosporiasis case-patients linked to the 2018 Temporospatial Cluster A (Epi Cluster Number [E.C.N] - 07), 2021 Florida Italian-style restaurant (E.C.N—24), 2020 Salad Chain A (E.C.N.–22), and 2019 Distributor A Type 3 (E.C.N.–16) epidemiologic clusters; isolates from these four distinct outbreaks were assigned to the same partition at a stringency of 94% (Fig 1). At optimal *s*-values, only isolates linked to the 2020 Salad Chain A (E.C.N.–22) and 2019 Distributor A (E.C.N.–16) epidemiological clusters remained in the same partition (Fig 2), supporting that these two outbreaks were caused by the same strain. In contrast isolates linked to 2018 Temporospatial Cluster A (E.C.N—07), and the 2021 FL Italian-style restaurant (E.C.N—24) cluster were divided across two distinct genetic partitions at higher stringency values (Fig 2). At stringency values above the established optima, some isolates were incorrectly separated from their epidemiologically linked partners (i.e., false negatives). For example, out of the 132 genotyped isolates linked to the pre-packaged salad mix 2020_001 cluster, all 132 belonged to a single genetic cluster at the optimal settings, while 11 isolates were split into other partitions at a stringency of 99.5% (Fig 3, S1 File). Likewise, 6 of the 94 specimens belonging to 2018 Vendor A split from their epi-linked partners at 99.5% stringency (Fig 3).

Simpson's index of diversity ranged from 0.9100 at *k* = 15 (*s*-values of 94% through 94.5%) to 0.9335 at *k* = 67 (*s* = 99.5%) (Table 2). At the optimal stringency values, *k* = 30 and Simpson's index of diversity was 0.9210, which is indicative of good discriminatory power.

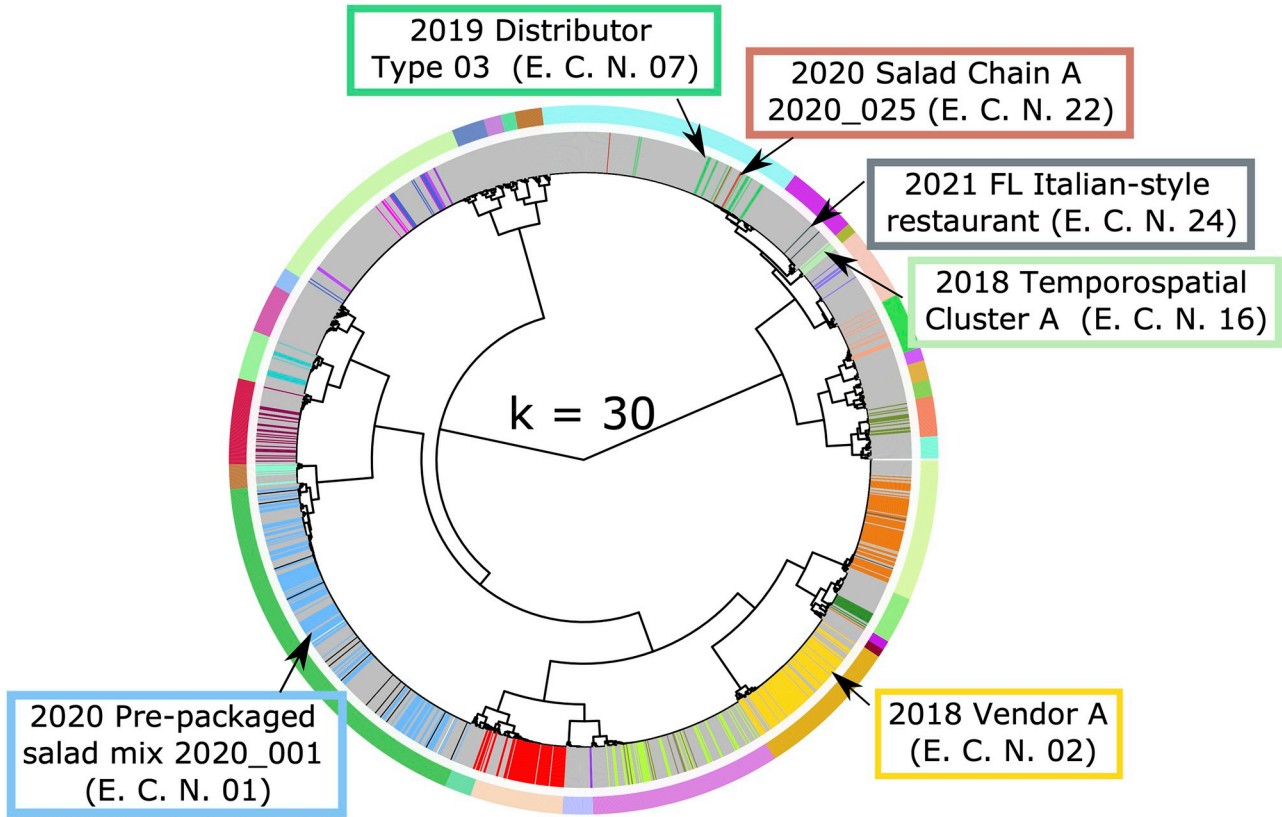

**Fig 2. Impact of optimal stringency (s) value on genetic linking of epidemiologically-linked isolates.** The hierarchical tree generated from our distance matrix was dissected into the optimal value of $k$ ($k = 30$) predicted using the 96.5% and 96.75% $s$-value. The outer circle of colored bars indicates the boundary between each partition predicted and the inner circle of colored bars represents the epidemiologic linkage of various isolates, where each bar color represents a distinct epidemiologic cluster (grey represents isolates of unknown epidemiologic linkage). Epidemiological clusters of interest are labeled in the colored boxes. (A) At $k = 30$, we observe that the labeled epidemiologic clusters on the top right of the tree are split between three different genetic partitions, while the two epidemiologic clusters on the bottom of the tree have 100% of isolates belonging to a single genetic partition.

## Discussion

We recently described a framework for dissecting hierarchically clustered genetic data that requires investigators to provide a user-defined stringency value that impacts downstream genetic partition memberships [4]. We recommended that the stringency parameter be set to a value between 95% and 100% and here we describe a process by which the selection of this parameter can be refined. While this seems like a small range of values, we demonstrate that even minor adjustments to the *s*-value can have a major impact on the resultant genetic partitions and subsequently, the perceived genetic linkages. This underpins the need for investigators to optimize user-defined parameters that impact the process of hierarchical tree dissection, regardless of the method employed.

Specifically, our results highlight the importance of selecting parameter values that maximize partitioning accuracy. In our investigation, all stringency values evaluated resulted in partitioning at an accuracy greater than 99%; however, at more relaxed stringency values (i.e., < 95%) greater than 70 false positives were observed, while at higher stringency values (i.e., > 98.5%) more than 30 false negatives were observed. At the optimal value established here, 0 false positives and only 6 false negatives were observed. Therefore, arbitrarily selecting a given

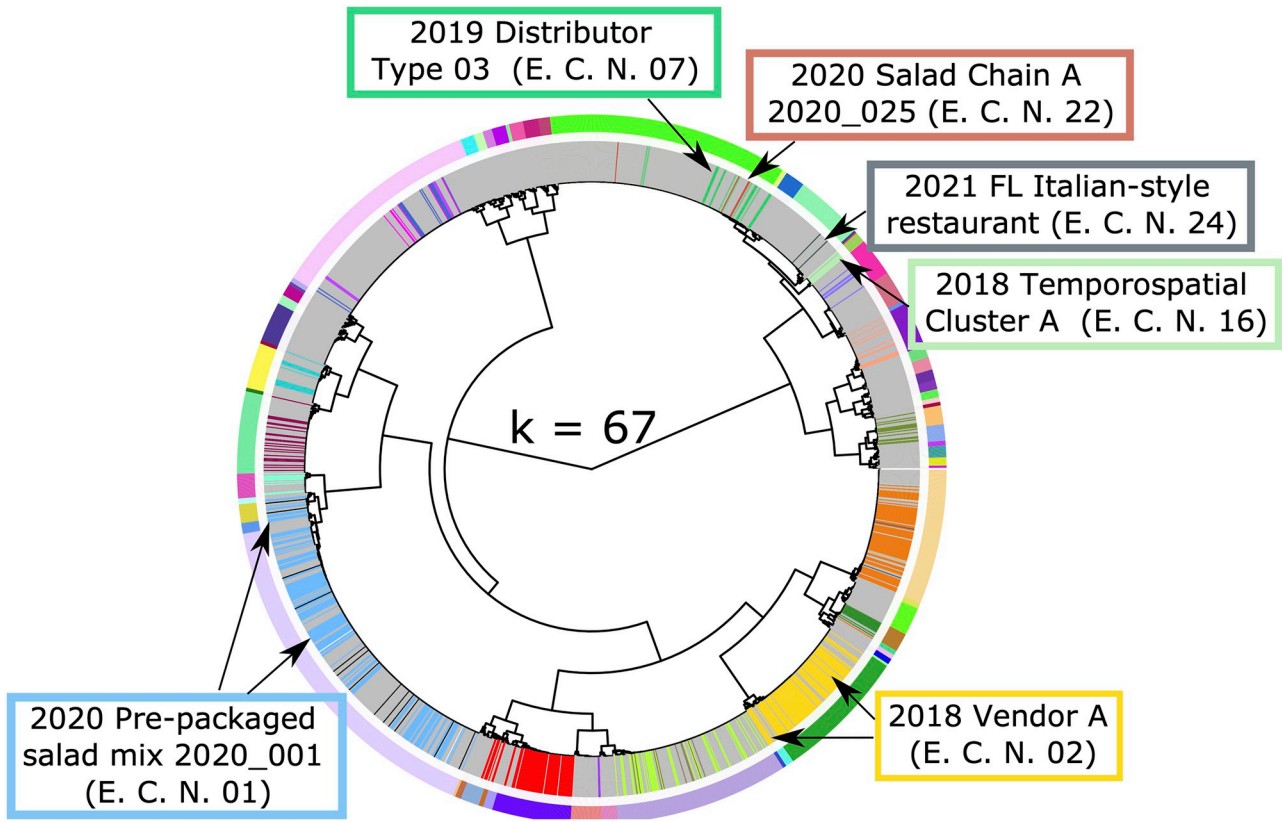

**Fig 3. Impact of maximum stringency (*s*) value on genetic linking of epidemiologically-linked isolates.** The hierarchical tree generated from our distance matrix was dissected into the optimal value of *k* (*k* = 67) predicted using the 99.5% *s*-value. The outer circle of colored bars indicates the boundary between each partition predicted and the inner circle of colored bars represents the epidemiologic linkage of various isolates, where each bar color represents a distinct epidemiologic cluster (grey represents isolates of unknown epidemiologic linkage). Epidemiological clusters of interest are labeled in the colored boxes. (A) At *k* = 99.5%, we observe that the labeled epidemiologic clusters on the bottom of the tree have isolates split across multiple genetic partitions, suggesting that the maximum stringency value is unnecessarily splitting epidemiologic clusters between partitions.

tree dissection parameter without a systematic evaluation across a range of potential values may result in a loss of performance by inflating the number of false positives or false negatives.

An important characteristic of any molecular epidemiologic tool is that the intersect between epidemiologically-defined clusters and their analogous genetic partitions should be nearing 100% [7]. Optimization of tree dissection parameters in this context should include computation of accuracy using epidemiologically defined clusters as a 'ground truth' reference for expected clustering outcomes. These reference genotypes of known epidemiologic linkage should be clustered alongside isolates of unknown epidemiologic linkage that represent possible candidates for downstream epidemiologic investigation. Resulting partitions identified using optimized parameter values that containing isolates with an unknown epidemiologic linkage will subsequently have a high likelihood of being derived from a common source, and thus represent robust candidates for epidemiologic follow-up. This is because their assignment to the same partition was based on parameters optimized to an internal reference of expected genetic links.

For epidemiologic investigations of cyclosporiasis outbreaks, patients complete a CNHGQ that collects information on the foods they recall consuming days to weeks prior to falling ill. Cyclosporiasis often presents as intermittent, non-specific symptoms many days to weeks after consumption of contaminated produce, meaning that weeks may elapse between illness onset

and CNHGQ interview. This delay can make it difficult for patients to recall specific meal components, potentially introducing noise to epidemiologic datasets [9]. Regardless, our experience with *Cyclospora* spp. has consistently demonstrated good concordance (~90% or more) between linkages identified via CNHGQ and genetic clustering. Accuracy of 100% may never be observed when assuming epidemiologic data as a true representative of 'ground truth', due to various sources of noise [9]. However, given the generally strong concordance between genotyping and these epidemiologic methods [8, 9], selecting parameter values that maximize accuracy seems warranted, as this–- in our experience—will usually yield partitions of unknown linkage that are more likely derived from a common source.

The discriminatory power of our dissected tree, determined by Simpson's index of diversity (*D*), was 0.9210 at the optimal *s*-values which resulted in 30 partitions. This is slightly lower than the value of *D* = 0.95 recommended elsewhere as an indicator of good discriminator power [7]. None of the stringency values evaluated here exceeded 0.95 (we observed a maximum *D* = 0.9335 at stringency = 99.5%), which is likely the result of a confluence of multiple factors. First, our dataset is heavily weighted to isolates from cyclosporiasis case patients residing in the United States (U.S.) between 2018 and 2021, which is unlikely to reflect the full genetic diversity of *Cyclospora* spp. (i.e., only genotypes causing U.S. infections during these periods were analyzed). Second, the current MLST-based genotyping approach captures only a portion of the ~45 megabase *Cyclospora* spp. genome [6]; the MLST method involves sequencing 8 genetic markers each less than 1 kilobase in length each. Finally, Simpson's index of diversity is a formula where datasets with greater richness (i.e., high number of clusters) and evenness (i.e., every cluster has a similar number of isolates) will have greater *D*-values compared to those with less richness and/or evenness. Our dataset consists of isolates from numerous cyclosporiasis outbreaks of varying sizes (Table 1), meaning richness and evenness are constrained by outbreak history, which impacts the value of *D*. Novel *Cyclospora* spp. types are identified each year [8, 9, 18] and work is being done to increase the number of markers used to genotype *Cyclospora* spp., thus discriminatory power will likely increase in response to these updates.

The sequencing of additional/different *Cyclospora* spp. MLST markers would warrant a reassessment of the optimal *s*-value, as subsequent tree structures may be impacted by the inclusion of the additional genetic information. Likewise, a large increase in the number of outbreaks caused by novel *Cyclospora* spp. genetic types, may also augment the resultant tree topology and thus be an impetus for re-evaluating the optimal value of the stringency parameter. Generally, when factors impact tree structure (e.g., new markers) or when the gold standard epidemiologic references do not encompass the observed genetic diversity (e.g., outbreaks from novel types) it is highly recommended that tree dissection parameters (i.e., SNP distance thresholds, or the stringency parameter in this case) be re-optimized. Nevertheless, the presently evaluated epidemiologic clusters represent the currently observed genetic diversity fairly well (Fig 1). Our optimal *s*-value (i.e., 96.5 to 96.75) was determined using a set of genotypes with gold-standard epidemiologic groupings, plus approximately 1,300 isolates without epidemiologic linkages. The optimal *s*-value described here remains a robust choice when applied to *Cyclospora* spp. that include the same gold-standard genotypes and a selection of the 1,300 additional isolates used here, in addition to any clinical isolates of interest to the investigator. A suggested reference dataset is provided (S2 File).

To conclude, we describe a simple approach that has proven useful for optimizing hierarchical tree dissection parameters to facilitate subsequent epidemiologic investigations. While the present example applies specifically to optimization of the stringency parameter for a particular tree dissection framework, this same approach could easily be used to optimize parameter values that are applicable to any tree dissection approach. We anticipate that other

molecular epidemiologists will find this work useful, especially in contexts where optimized parameters for tree dissection have not yet been established for certain pathogens.

## Supporting information

**S1 File. Complete clustering results.** This excel file contains a full list of calculation and results for accuracy and Simpson's $D$ at each $s$-value.
(XLSX)

**S2 File. Analysis support files.** This excel file includes the list of the suggested reference isolates, as well as the haplotype sheet and distance matrix used for clustering in this manuscript. The file also includes the epidemiologic linkages for each isolate.
(XLSX)

## Acknowledgments

We thank Yueli Zheng for bioinformatic support and Lauren Ahart and Marion Rice for assistance with epidemiologic classifications.

## Author Contributions

**Conceptualization:** Joel Barratt.

**Data curation:** David Jacobson.

**Formal analysis:** David Jacobson, Joel Barratt.

**Investigation:** David Jacobson, Joel Barratt.

**Validation:** David Jacobson.

**Visualization:** David Jacobson, Joel Barratt.

**Writing – original draft:** David Jacobson, Joel Barratt.

**Writing – review & editing:** David Jacobson, Joel Barratt.

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
