## [Decision Letter · Decision Letter 0]

19 Jan 2023

PONE-D-22-25671Optimizing hierarchical tree dissection parameters using historic epidemiologic data as ‘ground truth’PLOS ONE

Dear Dr. Jacobson,

Thank you for submitting your manuscript to PLOS ONE. After careful consideration, we feel that it has merit but does not fully meet PLOS ONE’s publication criteria as it currently stands. Therefore, we invite you to submit a revised version of the manuscript that addresses the points raised during the review process.

We look forward to receiving your revised manuscript.

Kind regards,

Nazanin Tajik

Academic Editor

PLOS ONE

and https://journals.plos.org/plosone/s/file?id=ba62/PLOSOne_formatting_sample_title_authors_affiliations.pdf.

Reviewers' comments:

Reviewer's Responses to Questions

**Comments to the Author**

1. Is the manuscript technically sound, and do the data support the conclusions?

Reviewer #1: Yes

Reviewer #2: Yes

2. Has the statistical analysis been performed appropriately and rigorously? 

Reviewer #1: Yes

Reviewer #2: Yes

3. Have the authors made all data underlying the findings in their manuscript fully available?

Reviewer #1: Yes

Reviewer #2: Yes

4. Is the manuscript presented in an intelligible fashion and written in standard English?

Reviewer #1: Yes

Reviewer #2: Yes

5. Review Comments to the Author

Reviewer #1: In this paper, the authors main research question is to optimize the hierarchical tree dissection parameter “stringency parameter” using historic epidemiologic data in order to improve the partitioning accuracy which represents the degree to which the tree partitions in this case genetic groupings of pathogens are similar to known epidemiologic groupings.

The authors hierarchically clustered 1,873 genotypes of the foodborne pathogen Cyclospora cayetanensis and dissected the tree using multiple values of the stringency parameter from 94% to 99.5% with increments of 0.25% in order to determine the optimal s value that maximizes the accuracy. The study shows that out of 23 different stringency values, clustering the C. cayetanensis isolates with known epidemiologic linkages using stringency parameters of 96.5% and 96.75% yield higher accuracy than values selected empirically.

The paper is well-written, and the results are helpful for interested readers of this journal. I believe that the article deserves publication. However, in my opinion, the following minor revision is needed.

Comments on the manuscript are as follows:

- Why did the author omit the Simpson index results and did not add to a table of results like Table 2 for example?

- The authors reported just the accuracy of the trees, why not add sensitivity specificity and precision as performance metrics as well to their results?

- The quality of the figure is very high, however in my opinion, the results can be better presented if each tree A, B and C are presented in separate figures.

Reviewer #2: 1. In table 2, various performance metrics such as sensitivity, specificity, PPV, and NPV can be calculated and compared.

2. Since choosing an appropriate distance statistic is fundamental, please explain why using Barratt’s heuristic definition of genetic distance is the best option.

3. It seems that references 16 and 22 are the same. Please omit one of them.

6. PLOS authors have the option to publish the peer review history of their article (what does this mean?). If published, this will include your full peer review and any attached files.

Reviewer #1: No

Reviewer #2: No

---

## [Author Response · Author response to Decision Letter 0]

23 Jan 2023

We appreciate the comments from the reviewers and we believe that we have addressed all of the comments listed below. Of interest, both reviewers suggested including additional performance metrics (e.g., specificity, precision, PPV) and this was a point of discussion between the authors before finalizing our initial submission. In our original submission, we ultimately decided to only include accuracy for the sake of brevity in the tables/text; however, it is clear that additional performance metrics would be useful, and we have now included sensitivity, specificity, PPV (i.e., precision), and NPV in our tables and results text in the revised manuscript. 

A further note is that our group has published an update regarding species naming for human-infecting Cyclospora parasites while this present manuscript was in review. We have updated our Cyclospora species naming convention in the revised manuscript to state Cyclospora spp. or Cyclospora sp., rather than Cyclospora cayetanensis. 

Please see our response to specific comments below.

Reviewer #1:

Comments on the manuscript are as follows:

1. - Why did the author omit the Simpson index results and did not add to a table of results like Table 2 for example?

a. Simpson index results for all stringency values have been added to Table 2.

2. - The authors reported just the accuracy of the trees, why not add sensitivity specificity and precision as performance metrics as well to their results?

a. Sensitivity, specificity, PPV, and NPV values have been added to Table 2 and the text results. Methods have been updated to mention calculation of these performance metrics.

3. - The quality of the figure is very high, however in my opinion, the results can be better presented if each tree A, B and C are presented in separate figures.

a. This is a good point and we agree that splitting the figure into separate figures makes for easier interpretation. As such, Figure 1 A-C has been split into Fig 1, Fig 2, and Fig 3, respectively. Manuscript text and figure legends have been updated appropriately.

Reviewer #2: 

1. In table 2, various performance metrics such as sensitivity, specificity, PPV, and NPV can be calculated and compared.

a. Sensitivity, specificity, PPV, and NPV values have been added to Table 2 and the text results. Methods have been updated to mention calculation of these performance metrics.

2. Since choosing an appropriate distance statistic is fundamental, please explain why using Barratt’s heuristic definition of genetic distance is the best option.

a. We have added text in the introduction section describing why Barratt’s heuristic was chosen to calculate genetic distances.

3. It seems that references 16 and 22 are the same. Please omit one of them.

a. Thank you for the attention to detail. This issue has been resolved and the initial citation #22 has been removed

---

## [Editor Report · Decision Letter 1]

8 Feb 2023

Optimizing hierarchical tree dissection parameters using historic epidemiologic data as ‘ground truth’

PONE-D-22-25671R1

Dear Dr. Jacobson,

We’re pleased to inform you that your manuscript has been judged scientifically suitable for publication and will be formally accepted for publication once it meets all outstanding technical requirements.

Kind regards,

Nazanin Tajik

Academic Editor

PLOS ONE
---

## [Editor Report · Acceptance letter]

14 Feb 2023

PONE-D-22-25671R1 

Optimizing hierarchical tree dissection parameters using historic epidemiologic data as ‘ground truth’ 

Dear Dr. Jacobson:

I'm pleased to inform you that your manuscript has been deemed suitable for publication in PLOS ONE. Congratulations! Your manuscript is now with our production department. 

Kind regards, 

on behalf of

Dr. Nazanin Tajik 

Academic Editor

PLOS ONE